# Artificial Intelligence in Capsule Endoscopy: A Practical Guide to Its Past and Future Challenges

**DOI:** 10.3390/diagnostics11091722

**Published:** 2021-09-20

**Authors:** Sang Hoon Kim, Yun Jeong Lim

**Affiliations:** Department of Internal Medicine, Dongguk University Ilsan Hospital, Dongguk University College of Medicine, Goyang 10326, Korea; spring0107@naver.com

**Keywords:** artificial intelligence, wireless capsule endoscopy, convolutional neural network, computer-aided reading, small bowel imaging

## Abstract

Artificial intelligence (AI) has revolutionized the medical diagnostic process of various diseases. Since the manual reading of capsule endoscopy videos is a time-intensive, error-prone process, computerized algorithms have been introduced to automate this process. Over the past decade, the evolution of convolutional neural network (CNN) enabled AI to detect multiple lesions simultaneously with increasing accuracy and sensitivity. Difficulty in validating CNN performance and unique characteristics of capsule endoscopy images make computer-aided reading systems in capsule endoscopy still on a preclinical level. Although AI technology can be used as an auxiliary second observer in capsule endoscopy, it is expected that in the near future, it will effectively reduce the reading time and ultimately become an independent, integrated reading system.

## 1. Introduction

The application and utilization of artificial intelligence (AI) in healthcare have been surprisingly fast. It is difficult for medical staff to properly acquire and analyze vast medical information within a limited time. In addition, due to the rapidly evolving diagnostic technology, its interpretation requires a higher degree of expertise than in the past.

Wireless capsule endoscopy (WCE) is expected to benefit the most among several endoscopic examinations from the development of pattern recognition AI technology. Reading a WCE case is a very tedious job, and the process is error-prone because it requires a long time and expertise. With the development of deep learning technology, research in this field is active, and it is very likely to be used in actual clinical practice in the future. However, even given the last decade of research, it has not yet been incorporated into real-world practice. This article provides state-of-the-art information about the current status of the technology, including the most widely used convolutional neural networks. Our article covers the historical evolution of CNN, binary classifications, automated classification of bowel cleansing, and novel real-time multi-lesion detecting systems that several existing reviews and meta-analyses on capsule endoscopy have not yet covered. Indeed, technological and regulatory hurdles that need to be overcome in the future are discussed.

## 2. The Rise of Artificial Intelligence

AI refers to artificially created intelligence, intelligence being a trait previously thought to occur only in humans. The concept involves computer programs that perform functions such as learning and problem-solving [1]. The method of enabling AI is called machine learning (ML) [2].

Samuel et al. defined ML as “a programming computer to learn from experience that should eventually eliminate the need for much of this detailed programming effort [1]”.

ML is roughly divided into supervised and unsupervised methods according to the training method. Supervised learning is a method of learning by labeled data, that is, the correct answer. In contrast, unsupervised methods are a method for the automated clustering of similar data based on commonalities. Sometimes, unsupervised learning is used as a primary method to identify appropriate features for subsequent supervised learning.

Among the ML methods, artificial neural networks (ANN) are a method inspired by the neuroanatomy of the brain. McCulloch and Pitts first proposed this concept in 1943 [3]. Similar to neurons in our brain, each neuron is a computing unit, and all neurons are connected, building a sophisticated network. After some time, a learning theory that could activate synapses between certain neurons and also be selectively strengthened emerged. Rosenblatt’s concept of perceptron devised a delta rule that modified the weight between each synapse [4]. The perceptron concept was evolved to learn non-linear functions by constructing multi-layered ANNs by Rumelhart and Hinton in 1986 [5]. A modernized multiple-layer perceptron (MLP) designed with input and output layers at both ends, which performed complex functions through numerous hidden layers between them, became popular.

MLP dramatically increased the non-linearity of the process and motivated researchers to solve every complex task. In 1980, Fukushima et al. proposed a pattern recognition method that recombined the features of the input pattern, which led to the induction of a convolutional neural network (CNN) [6]. In 2006, Hinton et al. named an MLP composed of several hidden layers, a deep neural network (DNN), and the DNN learning method was first called deep learning (DL) [7].

DL can be broadly classified into two categories. The first method is to pre-train the weight values by an unsupervised learning algorithm and then fine-tune them by the existing learning algorithm when the weight values get matured. The second method alleviates the problem of mixing error signals by limiting the number of connections between nodes in each layer. The former techniques include deep belief networks and a stacked auto-encoder, and the latter method is the CNN (convolutional neural network).

CNN models are being applied to image and medical data analysis in various areas. Automated image recognition in medical imaging has been investigated in several medical fields such as radiology, neurology, orthopedics, pathology, and gastroenterology.

## 3. Medical Imaging and CNN Model

Medical imaging, also known as diagnostic imaging, aids the physician in understanding the complications in a human body and enables them to make better decisions. Medical imaging is an extremely important element in medical practice today. It has allowed clinicians to learn more about the human body than ever before.

CNN has brought major changes in diagnostic image analysis, especially in the field of medical imaging. CNN can recognize patterns from images. Unlike previous machine learning, the algorithm is also responsible for its image feature extraction and performs the entire process up to image classification with a single model. With that characteristic, CNN is referred to as an end-to-end model [8]. As long as there is a desired image and an output to be classified, an algorithm with good performance can be easily implemented without hand-crafted features. In addition, it has a structure that can be used as an essential structure for detection and image segmentation beyond simple image classification, so it has become an essential technology for DL in the modern medical field.

The CNN model consists of three steps (Figure 1): (1) feature extraction, (2) feature dimension reduction, and (3) final classification. The first step, feature extraction, is performed with convolutional layers. During convolution, the kernel, an image filter of a certain size, scans the entire image, performs a weighted-sum operation, and delivers the output value to the next layer. In this process, three hyperparameters (depth, stride, and padding) are defined. Depth is how many kernels are used. Stride refers to the number of pixels that the kernel moves each time a convolution operation is performed. Padding is the application of a specific value to each edge of the input image so that the boundary information of the original image can contribute equally to feature extraction. The next step is called pooling, and it reduces the dimension of the feature. It does not utilize all features but selects only features that are effective for learning. The final step consists of MLP with the feature map (Figure 2) presented through the previous two feature extraction processes as input. This step is composed of fully connected layers, and the final classification (output) of an image can be derived.

## 4. Evolution of CNN

The winner of the ImageNet Large-Scale Visual Recognition Challenge (ILSVRC) competition in 2012, the AlexNet structure [9], increased the recognition rate of computer algorithms from 70% in the previous ten years to 85%. It was the first model to introduce a divergence function called ReLU (rectified linear unit), which solved the problems of weight adjustment due to the vanishing gradient issues during backpropagation for learning and dropout. In this way, while accurately normalizing the feature map, the contrast for strong features is increased. Afterward, GoogLeNet (2014) introduced the inception module that extracts various features from one layer by applying several kernels to one layer [10]. ResNet is the ILSVRC 2015 winning algorithm that surpassed human image recognition ability for the first time [11] (Figure 3). This network structure is seven times more deep than the previous GoogLeNet, so the learning performance is very sophisticated, and the weights are updated to sensitively detect even small differences in the input information. Recently, the Inception-ResNet v4 model maximized performance by combining with the inception module of GoogLeNet and using residual blocks [12].

## 5. State of the Art AI Studies in WCE

Since WCE was first introduced in 2001 [13], it has been a breakthrough in the diagnostic process of small intestine diseases. The average length of the small intestine in adults is about 600 cm. The range that can be observed with an endoscope is only a small part. There is no gold standard that anyone can agree on as a diagnostic tool for small intestine lesions. Various diagnostic methods, from the conventional balloon endoscopy to the recent motorized enteroscopy, are being used under the circumstances of each hospital. Balloon enteroscopy has the advantage of performing tissue biopsy or endoscopic treatment, but it requires a high degree of skill and much time. Recently, a motorized spiral Enteroscopy (PowerSpiral, Olympus, Tokyo, Japan) has been presented, enabling deeper exploration of the small bowel. Computed tomography (CT) enterography has limitations such as the requirement for contrast injection, radiation exposure, poor sensitivity for small tumors, and early inflammation. Magnetic resonance enterography (MR enterography) is a promising alternative, but it is too expensive to be used for screening in many parts of the world. In contrast, WCE has only a few contraindications, such as intestinal obstructions, patients having difficulty swallowing, and children. WCE is a tiny camera that can be swallowed and transmits consecutive captured images of digestive tract to a receiver. A capsule can pass throughout the rectum naturally within 72 h, but it can sometimes take up to weeks. It can sensitively recognize a lesion in a patient-friendly manner, so many clinical guidelines have suggested using it for several indications, including unexplained obscure gastrointestinal hemorrhage, small bowel Crohn’s disease, and small bowel neoplasms.

The interpretation and diagnosis of WCE images highly depend upon the human reader’s ability and require a time-consuming process. The CE for one case contains approximately 8–10 h of video, and about 50,000 to 70,000 pictures are obtained, in which only one or two lesions of interest may exist. As a result of the limitations of human concentration, the possibility of significant oversight is inevitable. To our knowledge, WCE readings have a significant miss rate of 5.9% for vascular lesions, 0.5% for ulcers, and 18.9% for neoplasms [14].

Due to the low consistency of the reading between physicians, WCE has been an ideal target of the AI system to assist physicians in identifying distinct lesions and areas of interest more easily [15]. Various commercial ML systems [16,17] have been commercialized, including Quick-View [18] (Medtronic, Minneapolis, USA), the suspected blood indicator [19] (Medtronic), and ExpressView [20] (Intromedic, Seoul, South Korea). However, they have little impact on reading and have not yet reached the stage of providing reliable classifications due to insufficient accuracy [21]. Here, we introduce CNN-based AI studies of various lesions that can be diagnosed by WCE (Table 1, Table 2).

### 5.1. Gastrointestinal Hemorrhage

The field of greatest interest in the early stage of research was automatic hemorrhage detection. It was first implemented using CNN (AlexNet) in 2016 by Xiao et al. [22]. They used 8200 training images and showed a sensitivity of 99.2% and a positive predictive value of 99.9%. Li et al. reported the results of a study comparing the performance and computational complexity of four CNN architectures (LeNet, AlexNet, GoogLeNet, and VGG-Net) in hemorrhage detection [23]. AlexNet seemed the most efficient system in terms of performance and computational complexity (sensitivity 98.9%). In addition, they reported that the performance related to images with low luminance or contrast was slightly poorer, and bubbles were highly susceptible to being falsely recognized as a hemorrhage. Other architectures such as MobileNet have been validated for bleeding image recognizing, resulting in comparable outcomes in spite of accuracy, recall, and F1 score [24].

### 5.2. Angioectasia

Angioectasia is a focal accumulation of dilated vessels in the mucosa and submucosa of the intestinal wall that may cause small intestinal bleeding. Leenhardt et al. reported the computer-aided detection of angioectasia (sensitivity 100%, specificity 96%) trained with 600 images in 2019 [25]. Since then, Tsuboi et al. showed a result of 98.8% sensitivity and specificity of 98.4% with a larger training image set and a richer validation image set (10,488 images) with a mixture of high-quality and imperfect images [26].

### 5.3. Erosion and Ulcers

Most WCE procedures are conducted to diagnose suspected Crohn’s disease of the small intestine, which is a disease that causes chronic inflammation and ulcerative changes in the intestinal mucosa. Therefore, much of AI research has been focused on this topic. However, the visual subtlety of ulcers makes them harder to discriminate from normal tissue than frankly red or actively bleeding lesions. Fan et al. achieved a sensitivity of 96.8% and specificity of 93.67% with AI training with 3250 ulcer and 4910 erosion images, which was the first use of DL to detect erosions and ulcers [27]. Aoki et al. conducted training and validation on a larger scale [28], and Wang et al. improved the efficiency of small ulcer detection through the so-called “second-glance detection framework” using multiple separate CNNs [29]. Another study by Aoki et al. quantified the clinical usefulness of the AI system by comparing its use with the standard clinical setting (manual reading), showing that their AI system reduced reading time (3.1 vs. 12.2 min) without reducing the detection rate (87% vs. 84%) while providing both classification confidence and a bounding box of the lesion area [30]. Recently, training and validation using 17,640 normal and lesion images of 49 patients with Crohn’s disease showed a high accuracy of 96.7% [31]. Interestingly, the median time required to obtain this impressive result was only 3.5 min per single small bowel case, which is only 10–25% of the manual reading time.

### 5.4. Celiac Disease

Computer-aided quantitative analysis of the existence and degree of celiac disease by a CNN-based DL model achieved 100% sensitivity and specificity for the external testing set [32]. It is found that the diagnostic probability of AI was also correlated to the severity level of small bowel mucosal lesions. However, the number of WCE cases used for training (11) and testing (10) was relatively small; it is believed that this algorithm should be used as a preliminary screening tool to select patients who need a following small bowel biopsy.

### 5.5. Polyps and Tumors

For detecting polyps and tumors, Yuan et al. introduced a stacked sparse autoencoder-based unsupervised approach, detecting polyps while also classifying normal images into turbid, bubble, and clear subtypes [33]. They achieved a sensitivity as high as 95.5% and a specificity of 98.5%. Saito et al. trained CNN using 5360 images with erosions and ulceration and validated it on 17,507 independent test set images (7507 protruding lesions with 10,000 normal images) [34]. According to each subtype (polyps, nodules, epithelial tumors, submucosal tumors, and venous structures), the reported sensitivity ranged from 86.5 to 95.8%.

### 5.6. Capsule Localization

The ability of AI to estimate the location of detected small bowel lesions is required for following a therapeutic approach. Conventionally, the reader records the passage time from the duodenum to the cecum and indirectly estimates the location using how long the lesion image was displayed after it passed the duodenum, but it lacks accuracy. Several ML approaches have been tried to overcome this problem since 2008 but have not been successful [35,36,37,38]. An in vitro study of WCE with unsupervised CNN measured both the traveled distance and the size of the detected lesions [39]. Although this was not a study applied within the human gastrointestinal tract, using an unaltered capsule in an artificial bowel showed a mean error of less than 0.01 cm in 20 cm of travel, showing the potential for this technology. There are opinions that several additional sensors (dual cameras, gyroscope, accelerometer, and magnetometer) should be introduced for capsule localization [40]. This can be solved only when the structural design of the capsule becomes more compact than the present design.

### 5.7. Automated Calculation of Bowel Preparation Quality

As the diagnostic yield of WCE highly depends upon the bowel preparation quality of passively obtained images, effective bowel cleansing is essential for qualified WCE examination. Accordingly, computed cleansing scores using the intensity of a tissue color bar (PillCam, Medtronic, Minneapolis, USA) or the map view (MiroCam, Intromedic, Seoul, South Korea) have also been developed. However, color intensity itself does not fully represent the cleanness of the entire WCE bowel on video. Therefore, Nam et al. developed a novel CNN-based (InceptionResnetV2) scoring system that calculates more objective, automated cleansing scores for small bowel preparation [41]. Such an AI system will enable a more objective assessment of the quality of WCE.

### 5.8. Binary Classification

Classifying the captured lesions in a binary manner (significant vs. insignificant) has attracted attention since it can diminish the workload of manual reading by sensitively selecting only the possible pathologic images that require manual reading. This system may be put into real-world practice faster than multi-lesion detection algorithms. Park et al. developed a practical Inception-ResNet based model that can detect various lesions and summarize WCE images binarily according to clinical significance [42]. By AI-assisted reading, lesion detection rates were improved (In experts; 34.3–73.0%; *p* = 0.029, In trainees; 24.7–53.1%; *p* = 0.029) while dramatically shortening the reading time (1621 vs. 746 min). Lui et al. used WCE images of the esophagus, stomach, and large intestine as well as the small intestine to classify anatomical landmarks and mucosal abnormalities in a binary way [43]. This binary system showed a sensitivity of 99.5% and a specificity of 98.5% in the small bowel.

### 5.9. Multiple Lesion Detection

Implementing an algorithm with a high level of accuracy for detecting multiple lesions is the ultimate goal of AI research. It requires a very large and balanced training dataset and detailed pinpoint annotations. Ding et al. collected 108 WCE cases from 77 medical centers [44]. They labeled and annotated over 158,000 images as normal and one of 10 abnormal categories (inflammation, ulcer, polyp, lymphangiectasia, bleeding, vascular disease, protruding lesion, lymphatic follicular hyperplasia, diverticulum, and parasites). In the performance comparison between the CNN and gastroenterologists using the validation images, the algorithm overpowered humans (sensitivity 99.88% vs. 74.57%, *p* < 0.001). The median time taken for the process was only 5.9 min, which was very short compared to the average manual reading time of 96.6 min. This result represents the potential of AI models for multiple lesion detection.
diagnostics-11-01722-t002_Table 2Table 2State of the art deep learning-based methods for wireless capsule endoscopy.
Study (Year)

Class

CNN Model

No. of Training/Validation Images

Accuracy

Sensitivity/Specificity
Xiao et al. (2016) [22]HemorrhageCNN8200/1800No info.99.2/No info.Li et al. (2017) [23]HemorrhageLeNetAlexNetGoogLeNetVGGNet9672/2418100.098.9/No info.Leenhardt et al. (2019) [25]AngioectasiaCNN600/600No info.100/96Tsuboi et al. (2020) [26]AngioectasiaSSD *2237/488No info.98.8/98.4Fan et al. (2018) [27]Erosion and ulcerAlexNetFor ulcers: 4400/3850For erosions: 5920/699095.296.8/94.8Aoki et al. (2019) [28]Erosion and ulcerSSD *5360/10,44090.888.2/90.9Klang et al. (2020) [31]Crohn’s diseaseCNN14,112/352896.796.8/96.6Yuan et al. (2017) [33]PolypsSSAE ^#^4000/No info.98.095.5/98.5Saito et al. (2020) [34]PolypsSSD *5360/17,507No info.86.5/No info.Ding et al. (2019) [44]Multiple lesionsResNet158,235/113,268,334No info.99.8 ^+^/100 ^+^SSD *, Single Shot MultiBox Detector; ^#^ SSAE, Stacked Sparse Autoencoder. ^+^ results of human CNN-based auxiliary reading.

## 6. Evaluating Clinical Performance of an Algorithm

Before using an AI algorithm on a patient, it is necessary to sufficiently validate how accurate it is and how helpful it is to clinical outcomes. Diagnostic errors may delay treatment for patients, resulting in serious harm to patients. In addition, the use of software that does not improve patient’s treatment outcome only increases unnecessary medical expense.

The development process of artificial intelligence algorithms consists of three stages: training, tuning, and testing. Among them, the tuning step is an optimization process that adjusts the algorithm’s hyperparameters, and it is not related to clinical validation. However, it is sometimes described as ‘validation’ in few papers, so caution is needed when reading.

Complex AI algorithms such as deep learning have a high degree of dependence on training data, so their accuracy is very high in the learning data. However, the accuracy is relatively low in the data not used for learning. This phenomenon is called ‘overfitting’, and various ‘regularization’ methods are used to reduce such problems. However, these methods also have limitations, so it is important to evaluate the performance using separate independent data not used for training and tuning, which is called ‘external validation’.

Ideal data for external validation has the following conditions: (1) representing the target clinical scenario/patients in real-world practice without remarkable bias, (2) collecting from multiple institutions other than those that contributed the training data, and (3) collecting prospectively. Without external validation, it cannot be generalized to actual clinical practice. There were many cases in which accuracy in the preclinical study was excellent and was difficult to use in real-world practice [45,46]. According to meta-analysis of 516 papers reporting the accuracy of machine learning algorithms, only 6% of studies evaluated the accuracy using external validation [47]. This shows that we need more attention and effort to verify the accuracy of the AI algorithm properly.

There are questions about which AI systems should be trusted more or less, and performing a comprehensive uncertainty quantification and trustworthiness is a very difficult process [48]. Well-known performance indicators for AI algorithms are accuracy, sensitivity, specificity, positive predictive value, negative predictive value, receiver operating characteristic curve, and precision–recall curve. These may be representative metrics, but they are still not perfect. Developing a format that can quantify the uncertainty of artificial intelligence that considers both the dataset and the trustworthiness of the inner algorithmic workings would be one of many future challenges.

## 7. Going beyond the Limits of Current Technology

Many studies have compared DL performance with existing handcrafted ML performance, and most studies in the field of WCE showed that DL outperformed ML [27,49]. This was because DL is an end-to-end learning system. In the case of ML, its limitations are obvious because it classifies with only manually selected features. However, in DL, the computing system itself performs feature selection for optimal results. This characteristic of DL is sometimes called a black box [50]. However, the need for a sufficiently large database is one of the limitations of DL-based approaches. While studying motility movement classification in WCE, Segui et al. reported that the accuracy improved only by 3% when the training data size increased ten-fold [49]. A well-known generative model is a generative adversarial network (GAN) that can collect a large amount of learning data. GAN is a structure first proposed by Ian Goodfellow in 2014 and shows great achievements in generating images, voices, and natural language [51]. For example, when there is a problem for a DL model to classify a dog or a cat by looking at a certain photo, the conventional algorithm has been taught to classify two groups by showing real dogs and cats. However, the trained model is very likely to make judgments based on only some characteristics of dogs and cats (e.g., pointed ears of cats, tails of dogs). However, suppose a model can generate pictures of dogs and cats rather than a simple classification model. In that case, it is believed that it better understands dogs and cats than the existing classification models. In the same vein as Richard Feynman said, “What I cannot create, I do not understand.”, generative models are very important. GAN is expected to be actively used to prepare data necessary for learning in a situation where it is difficult to collect a large amount of medical data required for AI learning due to the Data Security and Personal Information Protection Act. However, since the data used to train the generative model may also have serious weaknesses, it is essential to verify how generalizable it is in actual clinical situations.

An increasing number of CNN studies for WCE are being published not only in biocomputational but also in clinical journals. Studies from the biocomputational field are more prone to biases such as selection bias because important clinical information such as exclusion criteria is missing. As yet, studies that offer a solution to multiple gastrointestinal abnormalities are scarce. Most studies are limited to detecting a single type of lesion, which is insufficient for clinical practice. In addition, most studies extracted well-selected still images rather than using full-length videos. For images with low luminance, shaky, or low-image contrast, the problem of performance yield degradation is inevitable with the current technology. Overfitting is another crucial issue for DL. Although several approaches are used to mitigate overfitting such as cost function regularity, data augmentation, and relevant data selection, it remains an important problem for practical use [52]. Therefore, to overcome this, it is necessary to make an effort to build a large public dataset [53] and review and improve regulation policy suitable for proper investigations in each country. In addition, data imbalance with very few pathologic images during training needs to be addressed. In medical applications including classification of real-world capsule endoscopy videos, class imbalance is a common problem arising from the imbalance of positive and negative classes. If this phenomenon is serious from the training stage, the developed algorithm may run into problems with low recall of classes with small distributions. As a result, the program may perform well on the training set but underperform in predicting new unseen cases. This inherent limitation due to the nature of WCE in the small bowel calls for data augmentation technologies such as flip, crop, rotation, and blurring to be actively utilized. In addition, several sampling techniques including under sampling, up(over) sampling, and combined sampling have been introduced.

Additionally, current research based on retrospective studies is also prone to a high risk of investigator-induced bias. Future prospective multicenter CNN research on WCE is mandatory for real-world validation.

## 8. Conclusions

AI technology in WCE is still in a research phase that can only be experimentally used as an auxiliary second-observer in WCE. However, as image recognition architecture is evolving very fast, it is expected that shortly, it can effectively reduce the reader’s time, increase reading accuracy, and ultimately become a system that can independently achieve image reading. Prospective, multidisciplinary, multicenter WCE research is needed for clinical use in patient diagnosis.

## Figures and Tables

**Figure 1 diagnostics-11-01722-f001:**
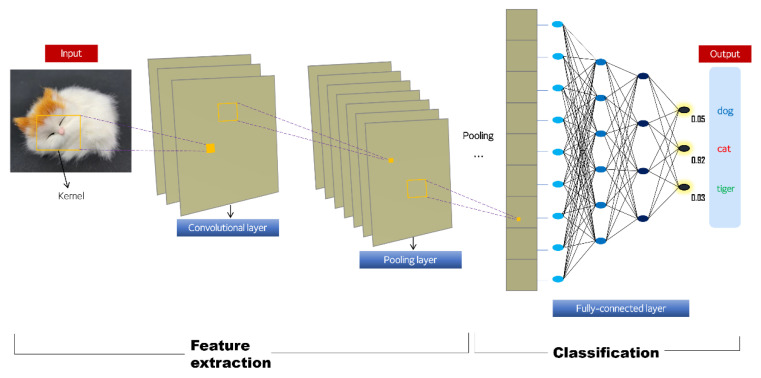
Schematic architecture of the convolutional neural network (CNN).

**Figure 2 diagnostics-11-01722-f002:**
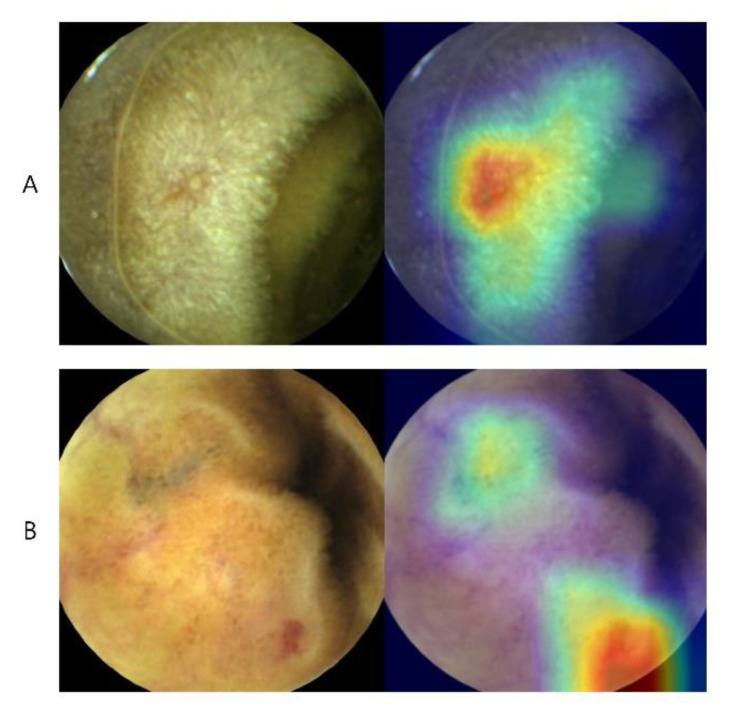
CNN’s class activation map (CAM) in capsule endoscopy image reading. This visualized feature map describes how the algorithm predicts each lesion. (**A**) Erosion with central depression of the mucosa is highlighted as red. (**B**) Simultaneous detection of both erosion and prominent vasculature.

**Figure 3 diagnostics-11-01722-f003:**
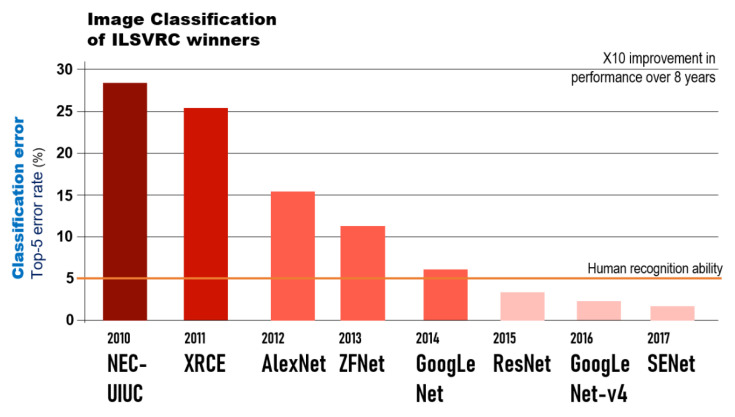
ILSVRC competition winning models by year and their rate of top-5 error (%).

**Table 1 diagnostics-11-01722-t001:** Research fields of deep learning-based methods for capsule endoscopy.

Gastrointestinal Hemorrhage (Bleeding)
Angioectasia
Erosion and ulcer (inflammation)
Celiac disease
Polyp and tumor
Capsule localization
Automated calculation of bowel preparation quality
Binary classification
Multiple lesion detection

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
