# Peer review of "Artificial Intelligence in Capsule Endoscopy: A Practical Guide to Its Past and Future Challenges"

_diagnostics, 2021, doi:10.3390/diagnostics11091722_

Round 1

Reviewer 1 Report

The article has reviewed the efficacy of artificial intelligence in capsule endoscopy that seems to be very promising as it may reduce the time spent on interpretation of images. I would recommend to describe the technical aspects of AI work in more acceptable manner for clinicians, perhaps the material should be illustrated.  Though the technology of AI in CE  is still in research phase it may be useful for clinicians for acquaintance. 

Reviewer 2 Report

Congratulations for the interesting work. Future publication should also detail te role of AI in colon capsule and crohn´s capsule endoscopy

Reviewer 3 Report

Kindly see the attached file for comments.

Reviewer 4 Report

Dear Editor, Dear Authors,

Thank you very much for the opportunity to read this manuscript.

The manuscript is well written in English and is pleasant to read. Its greatest advantage is the reference to current research on AI-assisted capsule endoscopy and its objectivity - without being overly optimistic, the Authors outlined the future of this technology. The greatest limitation of the manuscript is due to the limitations of the technology itself and the available research - we still know very little. Nevertheless, it is impossible not to agree with the Authors that AI is the future, but there is still much work (a collaborative effort) ahead of us.

Below, the Authors will find my detailed, albeit minor, comments on the manuscript.

  1. I suggest that the abbreviations should be standardized – WCE or CE.
  2. Line 26-27: Will benefit the most among endoscopic examinations or all diagnostic methods in medicine?
  3. Line 114: Invasive imaging methods of the small intestine, older (balloon endoscopy) and newer (e.g., PowerSpiral by Olympus) should also be considered here, with their pros and cons.
  4. Line 114: MRI enterography is nowadays used rather than CT, which lacks many of the limitations of CT. It is advisable to address this issue.
  5. Line 116: The use of the CE is also limited in children due to capsule size and swallowing problems.
  6. Line 117: This is not precisely a "swallowing disorder", but rather a problem with swallowing the capsule or the fear of swallowing it.
  7. Line 273: The abbreviation "ROC" is used only once, so it is unnecessary.

Round 2

Reviewer 3 Report

  1. Despite my comment and authors response that 'Abstract' is modified, I still see the same abstract in the 2nd version of the paper.
  2. The authors are advised to incorporate Comment 8 in the Introduction Section, it will define the scope of the current study and provide the context of the current study with respect to previous studies.
  3. I believe the references are in not proper font.

Author Response

Thank you very much for your comments.
